# Nrf2 Signaling Pathway as a Key to Treatment for Diabetic Dyslipidemia and Atherosclerosis

**DOI:** 10.3390/ijms25115831

**Published:** 2024-05-27

**Authors:** Michelle Yi, Arvin John Toribio, Yusuf Muhammad Salem, Michael Alexander, Antoney Ferrey, Lourdes Swentek, Ekamol Tantisattamo, Hirohito Ichii

**Affiliations:** 1Department of Surgery, University of California Irvine, Irvine, CA 92697, USA; myi10@hs.uci.edu (M.Y.); atoribi1@uci.edu (A.J.T.); ymsalem@uci.edu (Y.M.S.); michaela@hs.uci.edu (M.A.); lyrobles@hs.uci.edu (L.S.); 2Department of Medicine, University of California Irvine, Irvine, CA 92697, USA; ferreya@hs.uci.edu (A.F.); etantisa@hs.uci.edu (E.T.)

**Keywords:** diabetes mellitus, dyslipidemia, atherosclerosis, Nrf2 pathway, oxidative stress, antioxidant

## Abstract

Diabetes mellitus (DM) is a chronic endocrine disorder that affects more than 20 million people in the United States. DM-related complications affect multiple organ systems and are a significant cause of morbidity and mortality among people with DM. Of the numerous acute and chronic complications, atherosclerosis due to diabetic dyslipidemia is a condition that can lead to many life-threatening diseases, such as stroke, coronary artery disease, and myocardial infarction. The nuclear erythroid 2-related factor 2 (Nrf2) signaling pathway is an emerging antioxidative pathway and a promising target for the treatment of DM and its complications. This review aims to explore the Nrf2 pathway’s role in combating diabetic dyslipidemia. We will explore risk factors for diabetic dyslipidemia at a cellular level and aim to elucidate how the Nrf2 pathway becomes a potential therapeutic target for DM-related atherosclerosis.

## 1. Introduction

Diabetes mellitus (DM) is a chronic endocrine disorder characterized by dysregulation of blood glucose and insulin in the blood. Left untreated, DM causes various complications in different human organ systems of the human body. Of the numerous macrovascular and microvascular complications of DM, atherosclerosis, caused by diabetic dyslipidemia, is a condition that can lead to life-threatening diseases, including stroke and myocardial infarction (MI) [1]. Oxidative stress and chronic low-grade inflammation, some of the most significant pathogenic factors in diabetic comorbidities, become the main factors of DM-induced atherosclerosis [2,3,4,5]. In a study conducted on diabetic and non-diabetic human subjects, diabetic subjects had significantly higher coronary atherosclerotic parameters (i.e., incidences of coronary artery disease (CAD), obstructive CAD, calcified and non-calcified plaque, and mixed plaque) compared to non-diabetic subjects [6].

Atherosclerosis is an immunoinflammatory disease primarily in the intimal layer of the medium-sized and large arteries. It becomes a major underlying pathology of atherosclerotic cardiovascular diseases (ASCVDs) (Figure 1). Chronic accumulation of lipid and fiber plaques on arterial walls restricts blood flow and results in critical hypoxia in tissues [7]. Inflammation is initiated by the accumulation and modification of lipoproteins, such as low-density lipoprotein (LDL) [8]. The progression of the inflammatory lesion is then promoted by leukocyte recruitment and expression of pro-inflammatory cytokines. However, the main player in initial plaque formation in arteries is known to be altered lipid/lipoprotein profiles defined as dyslipidemia [7,9].

Dyslipidemia is a lipid disorder commonly characterized by increased cholesterol and triglycerides (TGs) as well as altered levels of associated lipoproteins such as LDL (especially its subtype small dense LDL (sdLDL)), very-low-density lipoprotein (VLDL), and high-density lipoprotein (HDL) [10]. Abnormal levels of various lipoproteins carrying different types of lipids become an important risk factor for ASCVD. Elevated plasma levels of LDL and LDL-cholesterol (LDL-C), cholesterol carried by LDL, are known to cause ASCVD by generating atherosclerotic plaques after undergoing oxidation and being engulfed by arterial wall macrophages, which then worsen the lesion by secreting pro-inflammatory cytokines [11,12]. sdLDL is a subtype of LDL that also becomes a major risk factor for atherogenic inflammation because it is susceptible to oxidative stress due to its decreased content of antioxidative vitamins [13]. VLDL, a TG-rich lipoprotein, is also known to be a contributor to plaque formation by binding to the macrophage scavenger receptors and VLDL receptors (VLDLRs) and lowering HDL levels [14,15,16]. Previous findings have highlighted HDL’s importance in cholesterol efflux from cells and anti-inflammatory effects on t-cell function [17,18], and low HDL levels also become a promoter of inflammation. Despite the substantial number of studies on the mechanisms of dyslipidemia and atherosclerosis, the extent to which hyperglycemia and oxidative stress contribute requires more thorough investigations.

The nuclear erythroid 2-related factor 2 (Nrf2) signaling pathway plays an important role in the prevention of various diseases. For instance, the activation of the Nrf2/Kelch-like erythroid cell-derived protein with CNC homology-associated protein 1 (Keap1) signaling was shown to induce the activation of the expression of NAD(P)H:quinone oxidoreductase 1 (NQO1), an antioxidant enzyme that protects cells against oxidative stress and inhibits the onset of ovarian cancer [19] and prostate cancer [20]. The activation of the Nrf2/Keap1 pathway also plays an important role in preventing non-carcinoma conditions such as preeclampsia [21] and metabolic-associated fatty liver disease [22].

As with other disorders involving oxidative stress, the Nrf2 signaling pathway is closely related to DM and atherosclerotic progression via combatting oxidative stress (Figure 2). Nrf2 is a molecule responsible for the activation of genes containing the antioxidant response element (ARE), an enhancer sequence, which then activates the transcription of antioxidant enzymes. When stimulated by oxidative stress, Nrf2 in the cytoplasm is detached from its suppressor, Keap1, and translocates into the nucleus to exert its gene-activating function [23]. Hyperglycemia, a condition that characterizes DM, activates the generation of reactive oxygen species (ROS) in mitochondria and angiotensin-II-mediated activation of NADPH oxidase 1 (NOX1) and NADPH oxidase 2 (NOX 2). NOX 1 and NOX 2, in turn, stimulate mitochondrial ROS and create a vicious cycle of oxidative stress [24,25]. Eventually, these oxidative stress factors become major contributors to various DM comorbidities by activating the nuclear transcription factor κB (NF-κB) pathway, which induces increased transcription of pro-inflammatory cytokines including tumor necrosis factor-alpha (TNF-α), transforming growth factor beta (TGF-β), and interleukin 1 (IL-1) [26,27]. The significance of effective Nrf2 nuclear translocation in inhibiting hyperglycemia-induced oxidative stress was shown in streptozotocin (STZ)-induced diabetic mice [28]. Once translocated, Nrf2 binds ARE found in the promoter region of the genes of various antioxidative enzymes that are related to DM-induced oxidative stress, including NQO1 [29], thioredoxin (Trx) [30], and heme oxygenase 1 (HO-1) [31]. In addition, Nrf2 suppresses the macrophage inflammatory response by inhibiting the transcription of proinflammatory cytokines by binding to the promoter regions of their genes [32]. Thus, enhancing the Nrf2 pathway can be a key to the cure of various oxidative-stress-induced diseases, like atherosclerosis. 

This review will explore antioxidative, Nrf2-involving treatments targeting different risk factors of diabetic dyslipidemia (e.g., VLDL, sdLDL, hepatic lipolysis-stimulated lipoprotein receptor (LSR) impairment), how each of the risk factors causes atherosclerosis, and how the treatments can therefore be used to treat atherosclerosis in patients with DM.

## 2. Diabetic Dyslipidemia and Atherosclerosis

It is important to note one of the main differences between the mechanisms of diabetic dyslipidemia and non-diabetic dyslipidemia includes insulin deficiency for type-1 diabetes mellitus (T1DM), or insulin resistance, in type-2 diabetes mellitus (T2DM) (Figure 3). The main mechanisms in DM causing dyslipidemia can be categorized into two types: increased activities of adipocyte and hepatic lipases and decreased activity of lipoprotein lipase (LPL) [33,34].

### 2.1. Lipase Upregulation

Increased activity of adipocyte lipase results in increased release of free fatty acids (FFAs) and TG from adipocytes to plasma. Hormone-sensitive lipase (HSL), one of the main adipocyte lipases, is directly regulated by insulin. Thus, insulin deficiency results in excessive lipolysis in adipocytes and increased plasma FFA levels by increasing the activity of HSL [35]. Excessive release of FFA becomes a risk factor for atherosclerosis not only directly but also indirectly by increasing hepatic VLDL production and hence TG [33,36]. Adipose triglyceride lipase (ATGL), another type of lipase in adipocytes, is also known to be increased with insulin deficiency [37]. The significance of the association between increased adipocytic lipolysis and insulin deficiency has been discussed and confirmed in previous studies [33,38]. 

The activity of hepatic lipase (HL) is increased under insulin-resistant conditions in T2DM [39,40,41]. Since HL plays an important role in converting TG into FFAs and altering the size and density of both LDL and HDL [33], unregulated HL activity can be a major exaggerating factor of dyslipidemia. Especially, regarding the alteration of LDL, HL hydrolyzes TG-rich remnants of VLDL and turns them into sdLDL [42], which is one of the major risk factors for diabetic atherosclerosis discussed later in this review. In addition, HL also interacts with intermediate-density lipoprotein (IDL) and converts it into cholesteryl-ester-rich LDL [43], which is one of the main cholesterol carriers in plasma and a risk factor for atherosclerosis.

### 2.2. LPL Downregulation

While upregulated lipase activities result in increased fatty acid release and altered lipoprotein profiles, downregulated LPL activities result in a deficiency in the clearance of TG-rich lipoproteins such as chylomicron (CM) and VLDL. In both T1DM and T2DM in humans, there is decreased LPL activity in post-heparin plasma [44], and such a change is associated with alterations in the lipoprotein pattern [45]. Since LPL plays an important role in VLDL metabolism by hydrolyzing its core TG, its inhibition can lead to hypertriglyceridemia and diminished postprandial clearance of TG-rich lipoproteins mainly due to increased apolipoprotein C-III (apoC-III), which interferes with the binding of apolipoprotein B (apoB) and apolipoprotein E (apoE) to hepatic receptors [14,46,47]. Thus, via LPL deficiency, impaired apoB-mediated lipoprotein clearance in the liver is eventually associated with high levels of VLDL and TG, both of which lead to atherosclerosis.

## 3. Highlighted Risk Factors of Diabetic Dyslipidemia and Atherosclerosis

### 3.1. VLDL

#### 3.1.1. How VLDL Is Formed in DM Environment

Along with CM, VLDL is a major TG carrier in plasma consisting of cholesterol, phospholipids, and various apolipoproteins (apoB-100, apoC, and apoE) besides TG [48]. Regulation of VLDL is important for the control of triglyceride levels and, thus, the prevention of atherosclerosis. Abnormally elevated plasma levels of VLDL followed by abnormally increased plasma TG concentration are observed mainly in patients with T2DM.

In the insulin-deficient environment of T1DM, a reduced insulin level decreases LPL activity, leading to impaired clearance of VLDL [48,49]. The correlation between insulin supply and LPL activity has been demonstrated, where LPL activity can be increased with insulin therapy in both rodents and insulin-dependent human subjects [50,51,52]. A study conducted on 16 insulin-deficient DM patients exhibited a significant negative correlation between LPL activity and untreated insulin deficiency. In addition to demonstrating increased VLDL and VLDL-TG levels, the results of the study also suggested the correlation between increased VLDL under insulin deficiency and LPL activity [53]. In a more recent study, VLDL levels were significantly lower in T1DM patients on intensified insulin therapy [54]. Another study conducted on insulin-treated T1DM patients also showed increased levels of non-HDL lipoproteins, including VLDL, to the extent that their lipoprotein profiles were better than those of non-diabetic controls [55]. Although it is often associated with insulin deficiency and T1DM, reduction in LPL or LPL activity is also observed in T2DM patients [56,57,58]. In summary, the results of the studies suggest the significance of insulin supply in VLDL control can further support the importance of optimized LPL activity in the maintenance of lipoprotein levels. 

VLDL can also be increased via increased hepatic apoB and apoC secretion, which are modulated by fatty acids. Excessive FFAs in the plasma entering the liver cause increased hepatic TG and apoC-III synthesis, which eventually leads to increased TG-rich VLDL in the plasma and hypertriglyceridemia [36]. Excess FFA also affects the level of apoB, an important protein for the VLDL assembly/secretion from the liver, by inhibiting its degradation [14,59,60]. Like the altered hepatic TG and apoC-III levels, this abnormality also leads to increased VLDL and TG levels in plasma [61]. Thus, atherosclerosis can be caused by not only FFA itself but also by increased apolipoproteins forming VLDL due to excess FFAs. As introduced in the previous section, upregulated HSL and HL activity in diabetic patients can cause increased plasma FFA. Altered HSL activity is seen in both T1DM and T2DM. In STZ-induced T1DM rats, 10 days of insulin deficiency significantly increased HSL expression and insulin therapy returned the impaired HSL activity to normal [62]. The mechanism of VLDL elevation in T2DM remains more multifaceted than that of T1DM. Since HSL is suppressed by insulin, several studies have demonstrated that HSL protein and/or activity in patients with T2DM were decreased [63] or unchanged [64] compared to the non-diabetic control group. This may be due to hyperinsulinemia in insulin-resistant states. Since insulin also directly governs hepatic apolipoprotein production, it has been suggested that VLDL is increased in insulin-resistant hyperinsulinemic obese patients because of resistance to the suppressive effect of insulin on hepatic apoB production [65]. Thus, the elevation of VLDL in T2DM is more likely to be due to impaired insulin action in the liver than HSL.

#### 3.1.2. Atherosclerosis and VLDL

VLDL increases the risk of atherosclerosis mainly through hypertriglyceridemia and endothelial inflammation. TG elevation is associated with delayed VLDL clearance from plasma and increased hepatic VLDL secretion [66]. Thus, both alteration of LPL and HSL activities explored in the previous section can become direct causes of hypertriglyceridemia. However, the exact mechanism by which TG causes atherosclerosis remains unclear. In addition, whether TG directly causes atherosclerosis by itself is still under debate. For instance, a study that involved patients with familial hyperchylomicronemia, a genetic disease accompanied by severe hypertriglyceridemia, found that these individuals did not develop atherosclerosis [67]. On the other hand, an observational cohort study conducted on individuals with low to moderate cardiovascular risk for the progression of early subclinical atherosclerosis noted that hypertriglyceridemia is significantly associated with subclinical atherosclerosis and vascular inflammation despite normal LDL-C levels [68]. 

It seems to be the inflammatory aspect of VLDL that has significance in atherosclerotic progression (Figure 4). As briefly introduced earlier in this review, VLDL and its remnants easily penetrate the arterial wall and can be taken up by scavenger receptors or VLDLRs of the arterial macrophages and form foam cells which become a major factor of initiation and aggravation of atherosclerosis [15,16]. Formation of foam cells then results in proinflammatory cytokine secretion, such as secretion of interleukin-1β (IL-1β) and interleukin-18 (IL-18), as well as enhancement of signaling via inflammatory receptors [69]. Despite the known theory, however, the scavenger receptor mechanism is under debate since a number of mouse studies have proven that deletion of scavenger receptors, such as the class A macrophage scavenger receptor (SR-A) or the cluster of differentiation 36 (CD36), does not reduce foam cell formation or atherosclerotic lesion formation in apoE-knockout mice [70,71]. Especially, regarding CD36’s ability to bind native lipoproteins like VLDL [72], these results suggest that the impact of scavenger receptors on atherogenesis is questionable. On the other hand, it has been shown that VLDLRs are expressed on macrophages in atheromatous plaques both in humans and rabbits [73]. Furthermore, a pro-atherogenic characteristic of VLDLR has been demonstrated by confirming the acceleration of atherosclerotic progression after transplantation of VLDLR^+/+^ macrophages into VLDLR^−/−^ mice [74]. Interestingly, there is a significant positive correlation between VLDLR levels and plasma triglyceride levels [75]. For planning antioxidative treatment targeting VLDL-involving atherogenesis, therefore, it would be more effective to approach the inflammation via VLDLRs rather than scavenger receptors.

Lastly, VLDL also aggravates atherosclerosis by decreasing the size and concentration of HDL, which has anti-inflammatory functions. A substantial number of studies have demonstrated the association between DM and depressed HDL levels [76,77,78]. It has also been shown that the size of HDL can be shifted toward small particles by hypertriglyceridemia [79], which as a consequence, results in high VLDL levels. Furthermore, recent studies have presented the finding that HDL particle size has a significant association with T2DM. These results show that patients with T2DM had lower concentrations of large HDL [80] or higher concentrations of small HDL [81] compared to non-diabetics. Furthermore, an alteration in HDL composition is also observed in individuals with T1DM, albeit the exact influence of the protein alteration on HDL function is yet to be discovered [82]. Nonetheless, it is still worth noting, since the altered structure of HDL is known to affect its antioxidant capacity [83]. Although the effect of TG itself on atherosclerosis is questionable, its indirect effect via lowering HDL levels is noteworthy. Besides HDL’s preventive feature against cholesterol efflux, it is worth focusing on the core of its antioxidant mechanism: paraoxonase 1 (PON1), an HDL-associated antioxidative enzyme. PON1 exerts its athero-protective function by reducing macrophage oxidative stress, monocyte recruitment, and LDL oxidation in atherosclerotic lesions [84,85]. Considering its impact on atherosclerosis, oxidative stress becomes a significant consequence of increased VLDL and TG.

#### 3.1.3. Nrf2-Targeting Treatments against VLDL-Dependent Oxidative Stress

Investigations have confirmed that the Nrf2 pathway is associated with oxidative stress derived from increased VLDLR expression by upregulating VLDLR expression. Such a link has been corroborated by verifying that VLDLR levels are increased by Nrf2 in the context of Peroxisome-Proliferator-Activated Receptor-β/δ (PPARβ/δ)-deficient hepatocytes in mice. According to the results of the study, an increased level of reactive oxygen species (ROS), induced by fructose, activates Nrf2 in hepatocytes, which then increases VLDLR levels [86]. 

There are several substances that are suggested to be regulators of VLDLR expression via the Nrf2 pathway (Table 1). The results of a study conducted on a Parkinson’s disease rat model have suggested that tauroursodeoxycholic acid (TUDCA), a bile acid derivative synthesized in hepatocytes, prevented ROS formation and lipid peroxidation by increasing the expressions of Nrf2 and its stabilizer, DJ-1 [87]. Additionally, in an obese mouse model, TUDCA was demonstrated to antagonize with Keap1, prevent Nrf2 degradation, and activate the Nrf2/ARE pathway [88].

Ethyl pyruvate (EP), an emerging anti-inflammatory substance derived from pyruvate, has also been studied and found to be an effective activator of Nrf2 and its translocation in astrocytes [89]. Interestingly, both TUDCA and EP are also known to have beneficial effects on DM. A substantial number of studies suggest that TUDCA improves both T1DM and T2DM mainly by reducing islet endoplasmic reticulum stress, insulin clearance, and beta cell dysfunction [90,91,92,93]. EP has also been studied in T1DM STZ mice for its effect on stimulating regulatory T-cells (Tregs), leading to a reduction in beta-cell loss [94]. Other studies also suggest its protective effect on various diabetic complications, such as liver injury [95], glomerular injury, and albuminuria [96], in STZ mice. Considering their inhibiting mechanisms against VLDLR expression and ameliorative effect on DM, TUDCA and EP can be promising Nrf2-targeting treatment options for diabetic atherosclerosis.

**Table 1 ijms-25-05831-t001:** Table of drugs/substances that can be used as treatments for both DM and atherosclerosis with their Nrf2-associated antioxidative actions: Nrf2, nuclear factor erythroid 2–related factor 2; VLDL, very-low-density lipoprotein; TUDCA, tauroursodeoxycholic acid; EP, ethyl pyruvate; ROS, reactive oxygen species; Keap1, Kelch-like erythroid cell-derived protein with CNC homology-associated protein 1; PON1, paraoxonase 1; sdLDL, small dense low-density lipoprotein; CDDO-Me, bardoxolone methyl (2-Cyano-3,12-dioxooleana-1,9(11)-dien-28-oic acid methyl ester); LSR, lipolysis-stimulated lipoprotein receptor; NF-κB, nuclear transcription factor κB.

Target Risk Factor	Drug/Substance	Action in the Nrf2 Pathway and Other Antioxidative Pathways	References
VLDL	TUDCA	Increases the expressions of Nrf2 and its stabilizer DJ-1	[87,88]
		Antagonizes with Keap1 to prevent Nrf2 degradation	
		Activates the Nrf2/ARE pathway	
		Alleviates lipid peroxidation and inflammatory response	
	EP	Activates Nrf2	[89]
		Activates Nrf2 translocation	
	Acacetin, curcumin	Enhances the Nrf2/Keap1 pathway	[97,98]
		Enhances PON1 concentration	
sdLDL	Metformin	Activates the Nrf2 pathway	[99]
		Prevents ROS generation	[100]
	CDDO-Me	Increases Nrf2 transcription and translation	[101]
		Binds to Keap1 and activates the Nrf2/Keap1 pathway	[102]
Hepatic LSR impairment	Calcitriol	Activates the Nrf2 pathway	[103]
		Impairs NF-κB	

The link between the Nrf2 pathway and PON1 has been suggested by identifying three putative Nrf2 binding sites on the promoter of the PON1 gene and confirming the enhancing effect of resveratrol on Nrf2 transactivation in PON1-transfected cells [104]. Such a discovery indicates that the Nrf2 pathway is a possible transcriptional regulator of PON1. The reverse relationship has been also explored; it has been found that overexpression of PON1 significantly mitigates acute dichlorvos poisoning and exerts antioxidative effects in mouse diaphragmatic muscle cells by inducing Nrf2 expression [105]. This relationship is not as thoroughly understood as that between Nrf2 and VLDLR expression, thus, the Nrf2 pathway and PON1 have an interesting positive correlation that requires deeper investigation.

Acacetin, a natural flavone commonly found in plant pigments, has been recently found to be an antioxidant that significantly enhances both the Nrf2/Keap1 pathway and PON1 concentration in atherosclerotic apoE-knockout mice [101]. Usage of curcumin and its degradation product 4-vinyl guaiacol demonstrate concurrent enhancement of Nrf2 and PON1. Findings suggest that hepatocytes treated with roasted curcumin and 4-vinyl guaiacol demonstrate increased Nrf2 transactivation and PON1 induction [102]. Although neither of the two studies explains the exact relationship or mechanism between the Nrf2 pathway and PON1, such findings are still worthy of attention considering their notable beneficial effects on DM and diabetic complications. Besides its anti-atherosclerotic feature, acacetin also exerts its antidiabetic effects by alleviating insulin resistance via regulation of Treg/T helper 17 (Th17) balance [106] and enhancing glucose uptake via the promotion of glucose transporter type 4 (GLUT4) translocation [107]. Likewise, curcumin is well-known for ameliorating insulin resistance and improving lipid profile in T2DM [108,109], as well as inhibiting diabetic cardiomyopathy via alleviation of oxidative stress [110]. Taken together, acacetin and curcumin are possible antioxidative treatments for diabetic atherosclerosis by linking the Nrf2 pathway and PON1 enhancement.

### 3.2. Small Dense Low-Density Lipoprotein (sdLDL)

#### 3.2.1. How sdLDL Is Formed in DM Environment

Studies have shown that sdLDL levels are high in patients who are prediabetic or diabetic [111]. The formation of sdLDL begins with the hydrolysis of TG and VLDL by LPL and HL, forming LDL. When the levels of triglycerides are above normal physiological conditions (>100 mg/dl fasting), the excess triglyceride and cholesteryl esters are exchanged by cholesteryl ester transfer protein, forming a higher level of LDL-TGs. At the same time, VLDL-TGs are hydrolyzed by LPL to form IDL. This IDL can be remodeled with hydrolysis of triglycerides by LPL and HL into further LDL. With the continuous exchange of triglyceride and cholesteryl esters forming LDL-TG, the triglyceride is finally hydrolyzed one final time by HL to form sdLDL [34]. Diabetic patients are typically at higher risk of developing hypertriglyceridemia [112]. This offers an explanation as to why diabetic patients may be prone to having higher levels of sdLDL, as the aforementioned LDL-triglyceride to sdLDL pathway will have higher activity because of elevated levels of triglyceride.

#### 3.2.2. Atherosclerosis and sdLDL

sdLDL is a subtype of LDL that many studies have substantiated as a biomarker for atherosclerosis. One study, which examined 11,419 human subjects’ serum sdLDL cholesterol levels, showed that elevated sdLDL was correlated to a higher risk of coronary artery disease [113]. Moreover, when assessing patients who were not considered at risk for cardiovascular disease, it has been shown that elevated levels of sdLDL cholesterol can independently account for these patients’ increased atherogenicity [114]. When it comes to diabetic patients, in a study analyzing the sdLDL cholesterol levels of 3684 T2DM patients, elevated levels of sdLDL cholesterol led to an increase in the severity of coronary heart disease (CHD). The study also found that these levels could be used to predict the likelihood of cardiovascular events [115]. Another study examined the overall size of LDL particles in T2DM patients and how the size affected overall coronary artery disease and atherosclerosis risk. It was found that the smaller-sized LDL correlated with a higher risk of CHD as well as an increase in intima media thickness [116]. With the evidence pointing to sdLDL as a biomarker for increased risk of atherosclerosis, it has been suggested that sdLDL levels should be screened in patients who have not been diagnosed with a risk for atherosclerosis or coronary artery disease through conventional clinical diagnostic tests [114].

There are many potential reasons that sdLDL causes a higher occurrence of atherosclerosis (Figure 2). Compared to more buoyant and larger LDL subtypes, sdLDL is more prone to oxidation [117]. It is hypothesized that, since sdLDL has less antioxidative and free cholesterol content as well as increased amounts of polyunsaturated fatty acids compared to larger, less dense LDL, sdLDL is much more likely to oxidize [13]. Oxidized low-density lipoprotein (oxLDL) has been shown mechanistically to directly lead to atherosclerosis. As found in a study conducted by Tani et al., the oxidation of LDL recruited monocytes into the intima, forming monocyte-derived macrophages. The study suggests that sdLDL is a significant promoting factor of lectin-like oxidized LDL receptor-1 (LOX-1), which becomes the receptor for oxidized LDL on macrophages and causes foam cell formation. Foam cells are then developed by the macrophages that take up oxLDL. These foam cells then form fatty streaks, which are characteristic of atherosclerosis [116]. Since sdLDL is the most likely subtype of LDL to oxidize, it can be concluded that sdLDL specifically is a main driving factor in developing atherosclerosis. The effects of oxidized sdLDL are further exacerbated by the fact that sdLDL has a longer circulation time than larger LDL subtypes due to a lower affinity for binding with the LDL receptor [117]. Another factor that increases the uptake of LDL by macrophages is the glycation of the apoB of LDL macrophages [117]. sdLDL was confirmed to be more susceptible to glycation than other LDL subtypes in a study conducted on 38 T2DM patients [117]. This study also further emphasized the risk of atherosclerosis in DM, as diabetic patients have higher levels of sdLDL and, thus, a greater risk for glycation or oxidation of sdLDL leading to atherosclerosis. Overall, it can be concluded that it is important for sdLDL levels to be lowered to prevent atherosclerosis, especially for diabetic patients.

#### 3.2.3. Nrf2-Targeting Treatments against sdLDL-Dependent Oxidative Stress

There are many potential treatment options for addressing sdLDL and its associated atherogenic risk. One approach tries to limit the level of sdLDL itself with drugs from the statin drug class, which are known to reduce overall cholesterol levels [118]. In a study on T2DM patients, atorvastatin 10 mg daily was given randomly to 1410 out of 2838 patients while the rest were given a placebo. The risks of cardiovascular events, acute CHD events, revascularization procedures, stroke, and death were significantly lower in the experimental groups compared to the placebo groups. It was also significant that no matter what the cholesterol levels of the patient were before the treatment, atorvastatin significantly improved outcomes across all ranges of initial cholesterol levels [119]. This treatment approach looks to eliminate the potential risks brought about directly by sdLDL. However, there is some evidence that the use of statins may worsen insulin resistance and blood glucose levels. In a retrospective cohort study examining 83,022 diabetic patients, there was a significant progression in DM with subjects who were prescribed statins. In this group, many patients needed to begin taking glucose-lowering medication because of worsening hyperglycemia and its related complications [120]. 

The other approach to decreasing the atherosclerotic susceptibility of diabetic patients is to use the Nrf2 pathway to help generate an antioxidative response from the body, to inhibit the uptake of oxidized LDL by arterial macrophages triggered by oxidized sdLDL (Table 1). Although this approach does not directly address sdLDL, it has the advantage of being able to prevent its associated risks more systematically. The drug metformin achieves this, as it has been shown to activate the Nrf2 pathway [99]. More importantly, it also lowers blood glucose levels. Metformin is one of the oldest effective DM drugs [121]. It has been well established that patients taking metformin for control of DM have a significantly reduced risk of developing atherosclerosis [122]. The mechanism by which this occurs was investigated in diabetic ApoE^−/−^/AMPK-α2^−/−^ and ApoE^−/−^ mice, where it was found that metformin allowed for the AMPK-mediated lowering of dynamin-related protein levels and preventing mitochondrial fragmentation [123]. While this study did not investigate the Nrf2 pathway, it did attribute the prevention of ROS generation as a factor for the reduction of atherosclerosis. Since metformin also activates the Nrf2 pathway leading to antioxidative products, the Nrf2 pathway may also help reduce sdLDL oxidation, but further investigation must be performed to understand if it does so with any significance.

Bardoxolone methyl (2-Cyano-3,12-dioxooleana-1,9(11)-dien-28-oic acid methyl ester, CDDO-Me), a synthetic triterpenoid, is another promising Nrf2 activator that has been studied for various diseases. CDDO-Me was revealed to increase both the transcription and translation of Nrf2 [101] in addition to activating the Nrf2/Keap1 pathway by binding to Keap1 [102]. CDDO-Me is a notable substance that has been widely studied for its effects on diabetic nephropathy. While CDDO-Me has often been suggested as a potential therapeutic agent for diabetic nephropathy [123,124,125], it is also noteworthy that CDDO-Me was also found to be ineffective in ameliorating end-stage renal disease in patients with T2DM with a higher rate of cardiovascular events occurring in the CDDO-Me patient group in a Bardoxolone Methyl Evaluation in Patients with Chronic Kidney Disease and Type 2 Diabetes Mellitus: the Occurrence of Renal Events (BEACON) trial [126]. While revealed to have effective aspects in the amelioration of diabetic nephropathy in several studies, whether CDDO-Me can be used as a safe therapeutic agent without major side effects for diabetic nephropathy is still in question. Interestingly, however, dh404, a derivative of CDDO-Me, was studied to ameliorate diabetes-induced atherosclerosis by inhibiting aortic oxidative stress and reducing LDL [127]. While its direct linkage to sdLDL is yet to be studied, the notable reduction in LDL by dh404 implies that its Nrf2-associated antioxidative effects can be a powerful therapeutic agent that can combat sdLDL increase and prevent DM-induced atherosclerosis.

### 3.3. Hepatic Lipolysis-Stimulated Lipoprotein Receptor Impairment

#### 3.3.1. How It Is Caused in DM Environment

The lipolysis-stimulated lipoprotein receptor (LSR) is a complex that can bind apoE or apoB once structural changes have been brought by the attachment of free fatty acids to the LSR, subsequently causing the breakdown of the mentioned lipoproteins once it is inside the complex. Impairment of LSR is linked to causing diseases such as cancer and atherosclerosis [128,129]. DM is no exception; in T1DM, a decrease of approximately 28% in LSR expression was seen [130] while, in T2DM, LSR function is dysregulated [128]. Reduced LSR expression is linked to the development of atherosclerosis in mice due to elevated levels of plasma non-HDL cholesterol as it affects apoB lipoprotein clearance [130]. Obstruction to the mechanism of insulin and hyperglycemia can potentially alter plasma lipoproteins in the DM environment. In T2DM, insulin resistance could cause irregularity in lipid formation. On the other hand, in T1DM, inadequate insulin can lead to a decrease in HDL production and an increase in triglyceride blood levels [131]. This highlights the importance of insulin regulation in both T1DM and T2DM [132]. Interestingly, abnormalities in types of lipids and lipid levels are similar in insulin-resistant environments, either T1DM or T2DM [133], again suggesting that dysregulation of insulin is associated with this defect. Along with insulin signaling, leptin signaling in mice affects LSR function. Leptin signaling has a significant role in regulating LSR protein levels, as seen in mice, as it helps increase the intake of lipids, significantly improving blood lipid levels [134].

Though multiple factors are associated with diabetic dyslipidemia, it has been shown in some studies that apoB production by the liver is elevated in T2DM patients, which is related to an increase in LDL levels and non-HDL-cholesterol levels, correlating to a higher risk of cardiovascular diseases (CVDs). apoB has a high chance of degeneration after protein translation due to the unregulated translation process. Moreover, the degeneration of apoB has been found to be amplified by insulin [135], suggesting that insulin has the capacity to influence the synthesis and production levels of other lipoproteins. For example, the excess production of VLDL results in lower HDL levels. This change in HDL composition increases the chance of small dense HDL (smHDL) forming or the breakdown of HDL [136]. To compensate for the increase in apoB levels, apoA generation is upregulated by ~25%, resulting in a 16% decrease in HDL concentration [137]. This compensatory action can limit the negative effects of increased apoB levels. 

#### 3.3.2. Atherosclerosis and Hepatic LSR Impairment

DM, especially T1DM, increases the risk of CHD [1,138]. Hyperglycemia plays a significant role in this process, as it expedites atherosclerosis due to changes in enzyme activity, increased oxidative stress, and stimulation of protein kinase C, which influences the expression of growth factors [139]. T1DM patients experience earlier onset of CHDs and atherosclerosis mainly as a result of hyperglycemia and oxidative stress leading to dyslipidemia and an increase in cholesterol levels [1,140]. Additionally, one pathway that can lead to DM-related CVDs is advanced glycation end-products (AGEs), which happens with elevated blood sugars. AGEs are difficult to break down and accumulate in long-term diabetic patients. AGE molecules can influence the transport of cholesterol molecules as they can reduce A1 and G1 adenosine-triphosphate (ATP)-binding transporters on monocytes. These adverse effects can lead to the development of CVDs. Moreover, AGE influences A1 and G1 ATP-binding transporters on monocytes and it reduces nitric oxide levels that can ultimately lead to reduced vasodilatory function. 

Dysregulation of the hepatic LSR affects the function of lipid level control in the blood. Single LSR gene knockout LSR −/+ mice model showed a 2-fold increase in plasma triglyceride levels compared to LSR +/+ mice models [141]. Note that elevated blood triglyceride levels are related to an increased mortality rate caused by CHD [142]. Interestingly, an LSR −/+ model resulted in significantly increased triglyceride levels ~2–3 h after diet administration compared to an LSR +/+ model, and the clearance rate of triglyceride was significantly (*p* < 0.04) lower in LSR −/+, leading to triglyceride levels being elevated for a longer time compared to LSR +/+ [142]. Impairment of LSR can be established to be linked to the increase in triglyceride levels that can lead to hypertriglyceridemia. The causality of hypertriglyceridemia in cardiovascular diseases is not yet established; however, there is evidence that lowered triglyceride levels caused by the LDL receptor gene lower the risk of CHDs via the reduction of apoB lipoproteins [143]. 

In T2DM mouse models, there is dysregulation in the function of LSR, and this can be confirmed through the inactivation of the LSR gene that leads to embryo death [144]. Moreover, impaired LSR function displayed negative impacts on triglyceride levels and processing [141]. Insufficient LSR function leads to reduced clearance of VLDL, which contains high levels of triglycerides and apoB lipoprotein post-meal, in a mice study [128]. Additionally, impaired LSR function caused irregular blood lipid profiles with significant highs of serum triglyceride levels and apoB/apoE lipoproteins [128]. The results of in vitro studies involving hepatocytes have indicated a significant function of LSR in lipoprotein intake [144,145], in which LSR dysfunction leads to a decreased lipoprotein intake, which can overall lead to LDL-C accumulation. All these factors combined can lead to a higher risk of CHDs, especially atherosclerosis, as it is associated with plaque accumulation. 

#### 3.3.3. Nrf2-Targeting Treatments against LSR-Impairment-Dependent Oxidative Stress

Leptin, a hormone released by adipose tissue, regulates multiple functions such as metabolism, energy homeostasis, and food intake, to name a few. It has been found that mutation of the LEP gene, the gene responsible for encoding the leptin hormone, is related to elevated fat mass and obesity [146,147]. In addition, a mutation in the gene responsible for encoding the leptin receptor is also linked to causing obesity [148]. Obesity can generate oxidative stress due to several reasons, such as hyperglycemia and mitochondrial dysfunction [149]. Leptin has been under investigation as a treatment option for obesity, further helping dyslipidemia and reducing the risks for atherosclerosis and other cardiovascular diseases. Furthermore, the administration of leptin in humans proceeded to result in significant weight loss [150] (Table 2). A study showed a mean weight loss of 7%, a reduced heart rate, and reduced inflammation of the heart, as assessed with the carotid-femoral pulse wave velocity and brachial-ankle pulse wave velocity [151]. This suggests that weight loss reduces the stiffness and inflammation of arteries, further lowering the risks associated with atherosclerosis and other CHDs. Interestingly, even healthy non-obese individuals can benefit. A ~10% loss of body weight resulted in positive changes in the lipid profile and blood pressure, and it limited inflammation in non-obese individuals [152]. The use of leptin to lower weight and reduce risks of atherosclerosis can be performed to improve the health outcomes of DM patients. 

More importantly, as mentioned in the previous section, leptin is known to be a promoting factor of hepatic lipid uptake via upregulation of hepatic LSR with increasing LSR protein and messenger ribonucleic acid (mRNA) levels in murine Hepa 1–6 cells. In agreement with previous findings on the negative effect of LSR impairment on lipid clearance, the murine study confirmed that LSR protein levels enhanced by leptin had significant effects on lipid clearance during the postprandial phase [134]. This suggests that leptin could be a treatment target for diminishing the oxidative stress caused by increased plasma TG and VLDL, which can then be approached using the Nrf2-involving treatments previously mentioned. The direct relationship between hepatic LSR impairment and the Nrf2 pathway is yet to be investigated.

Despite its positive effect on hepatic LSR activity and obesity, interestingly, leptin itself is also known to cause oxidative stress [153,156]. 1,25-Dihydrocycholecalciferol, such as calcitriol, an active metabolite of vitamin D, increases intestinal calcium and phosphate absorption [157]. It is used as a prescription to treat osteoporosis, hypocalcemia, and many other diseases. In T2DM, although calcitriol increased insulin secretion in the setting of vitamin D deficiency, calcitriol did not result in any changes regarding insulin resistance [158]. When used in patients with T1DM, calcitriol did not result in a significant alteration in blood glucose levels or glucose metabolism. However, it has been suggested that calcitriol might help because it has anti-inflammatory effects on the kidneys, further delaying diabetic nephropathy in T1DM patients [159]. 

Calcitriol has been shown to mitigate various complications, and interestingly, it also prevents oxidative stress caused by leptin molecules. Antioxidants are a necessary part of the human immune system when it comes to relieving oxidative stress. Oxidative stress can lead to a lot of degenerative pathways, such as complications in the heart and kidneys. The discovery of the use of calcitriol to increase vitamin D levels led to a significant decline in the production of superoxide anion in endothelial cells that were induced in leptin to undergo oxidative stress. This is significant as activation of the Nrf2 system was performed to inhibit oxidative stress, and this improved translocation of Nrf2 to the nucleus and impaired NF-κB [160].

In summary, treatment involving both leptin and calcitriol may reduce oxidative stress caused by impaired lipid clearance and obesity by upregulating hepatic LSR while limiting leptin-induced oxidative stress.

## 4. Conclusions

The Nrf2 signaling pathway is a promising therapeutic target for both DM and atherosclerosis. How the pathway can be used for the treatment of atherosclerosis in diabetic environments still requires further exploration and analysis. This review highlighted some of the major risk factors for diabetic atherosclerosis related to dyslipidemia, how they are caused in diabetic-specific conditions, such as insulin resistance or insulin deficiency, their mechanisms of causing atherosclerosis in relation to oxidative stress, and, most importantly, how and with what substances the risk factors can be approached and treated via the Nrf2 pathway. Several treatments have been well-studied for their efficacy in DM and its complications, yet their treatment mechanisms through the Nrf2 pathway or their possible connection to the pathway have not been well characterized. There are also several novel substances that have demonstrated antioxidative effects against DM-specific atherosclerotic risk factors through the Nrf2 pathway. Ultimately, the treatments explored in this review focus on Nrf2-dependent antioxidants that can be used as ameliorators of atherogenic oxidative stress caused by DM. In conclusion, this review suggests and reinforces the possible ability of the Nrf2 pathway as an effective remedy against diabetic dyslipidemia and atherosclerosis.

## Figures and Tables

**Figure 1 ijms-25-05831-f001:**
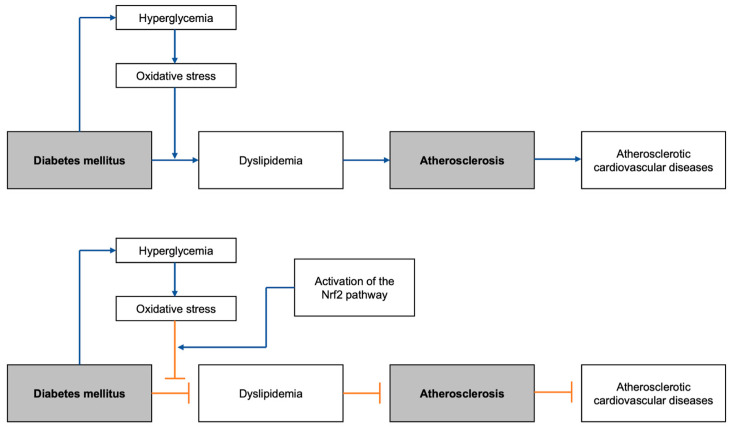
Graphical abstract summarizing the general flow of the development of DM-induced atherosclerotic cardiovascular diseases, with and without the Nrf2 pathway activation: Nrf2, nuclear erythroid 2-related factor 2.

**Figure 2 ijms-25-05831-f002:**
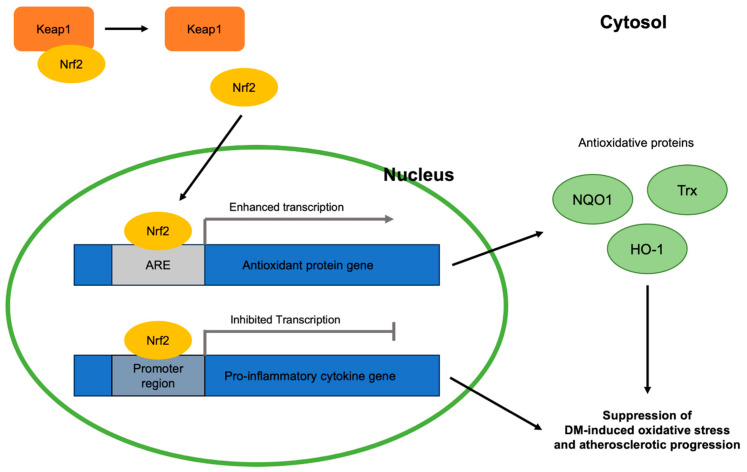
Antioxidative roles of Nrf2 in DM-induced oxidative stress and atherosclerotic progression viewed at the molecular level: Keap1, Kelch-like erythroid cell-derived protein with CNC homology-associated protein 1; Nrf2, nuclear erythroid 2-related factor 2; ARE, antioxidant response element; NQO1, NAD(P)H:quinone oxidoreductase 1; Trx, thioredoxin; HO-1, heme oxygenase 1; DM, diabetes mellitus.

**Figure 3 ijms-25-05831-f003:**
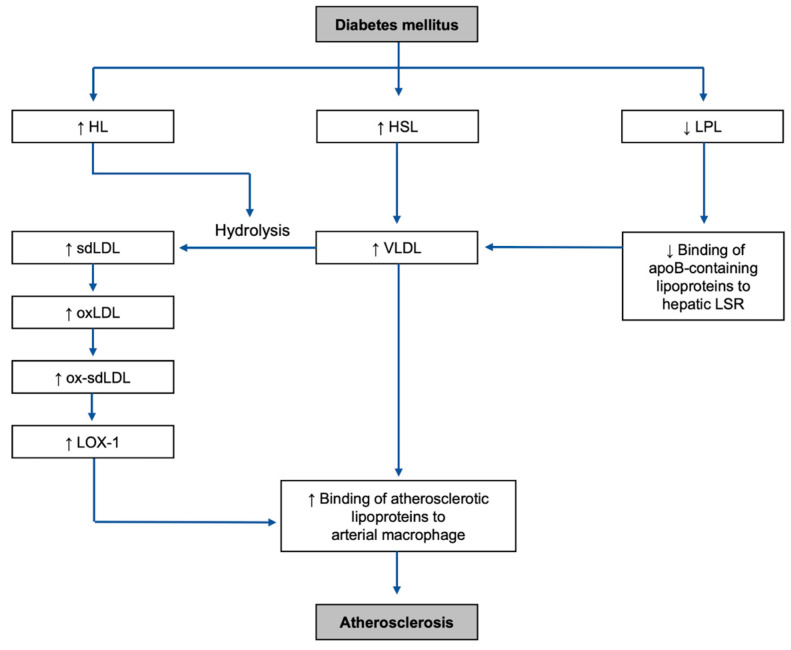
Overall mechanisms of the development of the highlighted risk factors and atherosclerosis in DM environment: HL, hepatic lipase; HSL, hormone-sensitive lipase; LPL, lipoprotein lipase; sdLDL, small dense low-density lipoprotein; VLDL, very-low-density lipoprotein; LSR, lipolysis-stimulated lipoprotein receptor; oxLDL, oxidized low-density lipoprotein; ox-sdLDL, oxidized small dense low-density lipoprotein; LOX-1, lectin-like oxidized LDL receptor-1.

**Figure 4 ijms-25-05831-f004:**
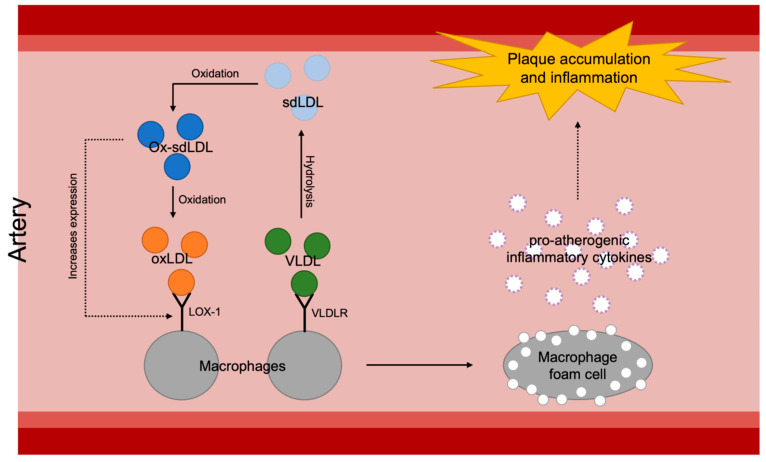
Overall mechanisms of plaque accumulation and inflammation of arterial walls caused by increased atherosclerotic lipoproteins: sdLDL, small dense low-density lipoprotein; ox-sdLDL, oxidized small dense low-density lipoprotein; oxLDL, oxidized low-density lipoprotein; VLDL, very-low-density lipoprotein; LOX-1, lectin-like oxidized LDL receptor-1; VLDLR, very-low-density lipoprotein.

**Table 2 ijms-25-05831-t002:** Table summarizing the results of the previous studies on the Nrf-targeting treatments against LSR-impairment-dependent oxidative stress: LSR, lipolysis-stimulated lipoprotein receptor; mRNA, messenger ribonucleic acid; T2DM, type-2 diabetes mellitus; T1DM, type-1 diabetes mellitus; Nrf2, nuclear erythroid 2-related factor 2; NF-κB, nuclear transcription factor κB.

Treatment	Subject	Study Results	References
Leptin	Human, obese	Significant weight loss	[150]
		Mean weight loss of 7%, reduced heart rate and heart inflammation	[151]
	Human, non-obese	10% (approximated) weight loss, positive changes in lipid profile and blood pressure, limitation of inflammation	[152]
	Mouse	Upregulation of hepatic LSR, increase of LSR protein and mRNA levels, lipid clearance during postprandial phase	[134]
	Human endothelial cells	Increased oxidative stress	[153]
Calcitriol	Human, T2DM	Increased insulin secretion under vitamin D deficiency without change in insulin resistance	[154]
	Human, T1DM	Alleviation of renal inflammation, delay of diabetic nephropathy, no significant alteration in the blood glucose levels or glucose metabolism	[155]
Leptin + Calcitriol	Human endothelial cells	Calcitriol-induced increase of vitamin D led to a decline in leptin-induced superoxide anion production (oxidative stress), improvement of Nrf2 translocation, impairment of NF-κB	[103]

## Data Availability

No new data were created or analyzed in this study. Data sharing is not applicable to this article.

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
