# Peer review of "Nrf2 Signaling Pathway as a Key to Treatment for Diabetic Dyslipidemia and Atherosclerosis"

_ijms, 2024, doi:10.3390/ijms25115831_

Round 1

Reviewer 1 Report

Comments and Suggestions for Authors

In this manuscript authors explored the role of NRF2 pathway in diabetic dyslipidemia illustrating the risk factors for diabetic dyslipidemia from a cellular level and elucidating how the NRf2 pathway could be a potential therapeutic target for DM-related atherosclerosis.

The manuscript is interesting and well written, English language is correct. Figures and Table are clear and easily readable. However, some points deserve to be improved.

Lines 25-30: It deserves to be pointed out that the vascular complications in DM are mainly due to an increased oxidative stress and a chronic low grade inflammation characterizing this pathology (see PMID: 37443812)

Lines 61-68: Since this is a review article, the multifunction role of NRF2/KEAP1 signaling pathway must be highlighted. In fact, this pathway is involved in the progression and onset of several cancerous and non-cancerous diseases (see PMID: 33123312; PMID: 37296665, PMID: 37525922, PMID: 36632321, PMID: 37175546).

3.3.3. Nrf2-targeting treatments against LSR-impairment-dependent oxidative stress: A table summarizing the results of the studies discussed in this section should be added

A figure summarizing the main role and modulation of NRF2/KEAP1 pathway in Diabetic Dyslipidemia and Atherosclerosis should be added

Please, uniform NRF2 throughout the text

Abbreviations must be written in full length when mentioned for the first time

A graphical abstract would be helpful

Author Response

Reviewer #1

In this manuscript authors explored the role of NRF2 pathway in diabetic dyslipidemia illustrating the risk factors for diabetic dyslipidemia from a cellular level and elucidating how the NRf2 pathway could be a potential therapeutic target for DM-related atherosclerosis.

The manuscript is interesting and well written, English language is correct. Figures and Table are clear and easily readable. However, some points deserve to be improved.

Lines 25-30: It deserves to be pointed out that the vascular complications in DM are mainly due to an increased oxidative stress and a chronic low grade inflammation characterizing this pathology (see PMID: 37443812)

    Thank you for the comment and the recommended reference. We added an explanation about the link between diabetic atherosclerosis and diabetes-induced oxidative stress/inflammation. We also added the reference you suggested to explain this part (lines 31-33).

Lines 61-68: Since this is a review article, the multifunction role of NRF2/KEAP1 signaling pathway must be highlighted. In fact, this pathway is involved in the progression and onset of several cancerous and non-cancerous diseases (see PMID: 33123312; PMID: 37296665, PMID: 37525922, PMID: 36632321, PMID: 37175546).

    We appreciate the suggestion. We added an explanation about the multifaceted role of the Nrf2/Keap1 pathway in both cancerous and non-cancerous conditions using the suggested references (lines 69-76).

3.3.3. Nrf2-targeting treatments against LSR-impairment-dependent oxidative stress: A table summarizing the results of the studies discussed in this section should be added

    Thank you for the suggestion. We added Table 2, which summarizes the study results mentioned in Section 3.3.3 (lines 507-510).

A figure summarizing the main role and modulation of NRF2/KEAP1 pathway in Diabetic Dyslipidemia and Atherosclerosis should be added

    Thank you for the feedback. We added a diagram that visualizes the role of the Nrf2/Keap1 pathway in ameliorating DM-induced oxidative stress and atherosclerosis (Figure 2, lines 101-106).

Please, uniform NRF2 throughout the text
    Thank you for the heads-up. We uniformed Nrf2 throughout the text by fixing two Nrf2 abbreviations (lines 19 and 20).

Abbreviations must be written in full length when mentioned for the first time
    Thank you for the feedback. We expanded the abbreviations that needed to be addressed (lines 35, 219, 222-223, 313, 355-356, 460, and 514).

A graphical abstract would be helpful

    Thank you for your input. We added a graphical abstract that summarizes the general flow of the atherosclerotic development discussed in this review (Figure 1, lines 47-50)

Reviewer 2 Report

Comments and Suggestions for Authors

In my opinion, the work is interesting.

I.                 However, it would be worth expanding in the introduction on the multidirectional action of the Nrf2 signaling pathway in the anti-inflammatory context, which also has an impact on diabetes. And emphasize the role of influence on the development of atherosclerosis

For example

11.      The recent advancements in elucidation of ARE/keap1/Nrf2 pathway can help in better understanding of diabetes mellitus. Various clinical trials and animal studies have shown the promising effect of Nrf2 pathway in reversing diabetes by counteracting with the oxidative stress produced. The gene is known to dissociate from Keap1 on coming in contact with such stresses to show preventive and prognosis effect. The Nrf2 gene has been marked as a molecular player in dealing with wide intracellular as well as extracellular cellular interactions in different diseases. The regulation of this gene gives some transcription factor that contain antioxidant response elements (ARE) in their promoter region and thus are responsible for encoding certain proteins involved in regulation of metabolic and detoxifying enzymes.

22.      Nrf2 presents promising role not only in diabetes mellitus, but also in its associated complications. The generation of ROS via mitochondria and the persuaded commencement of NOX generating angiotensin II leads to hyperglycaemia. These factors further cause activation of NFκB pathway that causes transcription of infammatory and fbrotic genes like TNF-α, TGF-β and MCP1, resulting into diabetes mellitus. However, it shall be noticed that under normal conditions Keap1, which acts as negative regulator of Nrf2, targets it constitutively for its degradation and ubiquitination in cytosol. Conversely, the stabilization of Nrf2 is dependent mainly on oxidative stress induced by hyperglycaemia, ultimately releasing Keap1 and allowing translocation of Nrf2 into nucleus in order to enable it to activate ARE related genes like CAT, SOD, HO-1 and NQO. This reduces the levels of ROS by cytoprotective and antioxidant action, resulting into reduction of diabetes mellitus

33.      Modifying effect on pro-inflammatory cytokines

II.                Please expand the CAD abbreviation on line 32

Author Response

Reviewer #2

In my opinion, the work is interesting.
I.                 However, it would be worth expanding in the introduction on the multidirectional action of the Nrf2 signaling pathway in the anti-inflammatory context, which also has an impact on diabetes. And emphasize the role of influence on the development of atherosclerosis
For example
11.      The recent advancements in elucidation of ARE/keap1/Nrf2 pathway can help in better understanding of diabetes mellitus. Various clinical trials and animal studies have shown the promising effect of Nrf2 pathway in reversing diabetes by counteracting with the oxidative stress produced. The gene is known to dissociate from Keap1 on coming in contact with such stresses to show preventive and prognosis effect. The Nrf2 gene has been marked as a molecular player in dealing with wide intracellular as well as extracellular cellular interactions in different diseases. The regulation of this gene gives some transcription factor that contain antioxidant response elements (ARE) in their promoter region and thus are responsible for encoding certain proteins involved in regulation of metabolic and detoxifying enzymes.
22.      Nrf2 presents promising role not only in diabetes mellitus, but also in its associated complications. The generation of ROS via mitochondria and the persuaded commencement of NOX generating angiotensin II leads to hyperglycaemia. These factors further cause activation of NFκB pathway that causes transcription of infammatory and fbrotic genes like TNF-α, TGF-β and MCP1, resulting into diabetes mellitus. However, it shall be noticed that under normal conditions Keap1, which acts as negative regulator of Nrf2, targets it constitutively for its degradation and ubiquitination in cytosol. Conversely, the stabilization of Nrf2 is dependent mainly on oxidative stress induced by hyperglycaemia, ultimately releasing Keap1 and allowing translocation of Nrf2 into nucleus in order to enable it to activate ARE related genes like CAT, SOD, HO-1 and NQO. This reduces the levels of ROS by cytoprotective and antioxidant action, resulting into reduction of diabetes mellitus
33.      Modifying effect on pro-inflammatory cytokines
    Thank you for such insightful suggestions. All of the three examples you suggested have been reflected in our current draft with in-depth explanations (lines 78 and 83-97). 

II.                Please expand the CAD abbreviation on line 32
    Thank you for the feedback. We expanded the CAD abbreviation (line 35). 

Reviewer 3 Report

Comments and Suggestions for Authors

This narrative review focuses on dyslipidemia, diabetes, oxidative stress, and Nrf2 signaling in the context of atherosclerosis. While the topic is interesting, there are several concerns:

1-The Nrf2 signaling pathway requires a more comprehensive description. I suggest including a figure detailing the molecular pathway.

2-There appears to be some inconsistency that needs further discussion, as benefits in atherosclerosis have been described for both inhibiting the Nrf2 pathway (TUDCA and EP) and enhancing it (acacetin, curcumin, metformin, or calcitriol).

3-Bardoxolone, a notable compound that affects Nrf2 signaling, was not mentioned.

Author Response

Reviewer #3

This narrative review focuses on dyslipidemia, diabetes, oxidative stress, and Nrf2 signaling in the context of atherosclerosis. While the topic is interesting, there are several concerns:
1-The Nrf2 signaling pathway requires a more comprehensive description. I suggest including a figure detailing the molecular pathway.

    Thank you for the input. We added a diagram that visualizes the role of the Nrf2/Keap1 pathway in ameliorating DM-induced oxidative stress and atherosclerosis (Figure 2, lines 101-106).
2-There appears to be some inconsistency that needs further discussion, as benefits in atherosclerosis have been described for both inhibiting the Nrf2 pathway (TUDCA and EP) and enhancing it (acacetin, curcumin, metformin, or calcitriol).
    Thank you for the feedback. Through further research, we found previous studies that demonstrate the Nrf2-activating effects of TUDCA and EP. To avoid any confusion, we included these studies instead of the one that had been included in the previous draft (lines 269-277, Table 1).
3-Bardoxolone, a notable compound that affects Nrf2 signaling, was not mentioned.
    Thank you for the suggestion. Bardoxolone methyl (CDDO-Me) has been added to the sdLDL section with a thorough explanation (lines 407-417, Table 1). 

Round 2

Reviewer 1 Report

Comments and Suggestions for Authors

Manuscript has been improved and can be accepted for publication.

Author Response

Thanks you so much. 

Reviewer 3 Report

Comments and Suggestions for Authors

Please do not use multiple colors to mark changes in the manuscript (some in pink, others in yellow, blue, green, etc), as it is difficult to read. Additionally, I still have concerns about the focus which seems more centered on dyslipidemia and atherosclerosis than on NRF2

1-I Couldn’t find the figure that illustrates the role of the Nrf2/Keap1 pathway in ameliorating DM-induced oxidative stress and atherosclerosis. Current Figure 2 was focused on the mechanisms of plaque accumulation and arterial wall due to increased atherogenic lipoproteins, rather than on the NRF2/keap1 pathway.

2-Bardoxolone needs more discussion. References 125-127 were devoted to leptins rather than bardoxolone. In addition, it was stated that bardoxolone “ is also often suggested as a potent therapeutic agent for diabetic nephropathy”. However, the BACON TRIAL ( 10.1056/NEJMoa1306033), bardoxolone did not reduce the risk of End Stage Renal Diseases or death from cardiovascular causes in type 2 diabetes mellitus and stage 4 chronic kidney disease. A higher rate of cardiovascular events with bardoxolone than with placebo prompted the termination of the trial. However, this negative result was not mentioned in the review, suggesting a bias toward positive results, which is a significant flaw.

Author Response

Please do not use multiple colors to mark changes in the manuscript (some in pink, others in yellow, blue, green, etc), as it is difficult to read. Additionally, I still have concerns about the focus which seems more centered on dyslipidemia and atherosclerosis than on NRF2

  • Thank you for the feedback. Highlight colors have been unified into one color. Regarding the expansion of Nrf2 coverage in the review, we added a detailed discussion about the role of Nrf2 in a hyperglycemic environment and hyperglycemia-induced oxidative stress. In addition, a detailed explanation of the role of Nrf2 in both cancerous and non-cancerous diseases besides diabetes has been added (lines 69-76).

1-I Couldn’t find the figure that illustrates the role of the Nrf2/Keap1 pathway in ameliorating DM-induced oxidative stress and atherosclerosis. Current Figure 2 was focused on the mechanisms of plaque accumulation and arterial wall due to increased atherogenic lipoproteins, rather than on the NRF2/keap1 pathway.

  • Thank you for the input. There was some confusion during the manuscript file submission, and a version without new figures was submitted instead of the final version. Please refer to Figure 2 in the new draft we are submitting this time. Figure 2 focuses on the antioxidative roles of the Nrf2/Keap1 pathway, viewed at a molecular level (lines 101-106). We apologize for the confusion.

2-Bardoxolone needs more discussion. References 125-127 were devoted to leptins rather than bardoxolone. In addition, it was stated that bardoxolone “ is also often suggested as a potent therapeutic agent for diabetic nephropathy”. However, the BACON TRIAL ( 10.1056/NEJMoa1306033), bardoxolone did not reduce the risk of End Stage Renal Diseases or death from cardiovascular causes in type 2 diabetes mellitus and stage 4 chronic kidney disease. A higher rate of cardiovascular events with bardoxolone than with placebo prompted the termination of the trial. However, this negative result was not mentioned in the review, suggesting a bias toward positive results, which is a significant flaw.

  • Thank you for the heads-up. We acknowledge that it is indeed important to mention the downside of bardoxolone methyl in diabetic nephropathy. We mentioned the results of the BEACON trial you suggested, in addition to clarifying the ambivalence of bardoxolone methyl in diabetic nephropathy (lines 411-420).

Round 3

Reviewer 3 Report

Comments and Suggestions for Authors

I agree with the authors' response and the paper has been improved.